# Reproducing Meta-learning with differentiable closed-form solvers

**Arnout Devos**[*]**, Sylvain Chatel**[*]**, Matthias Grossglauser**
Department of Computer and Communication Sciences
Swiss Federal Institute of Technology Lausanne (EPFL)
`{first.last}@epfl.ch`

## Abstract

In this paper, we present a reproduction of the paper of Bertinetto et al. [2019] *"Meta-learning with differentiable closed-form solvers"* as part of the ICLR 2019 Reproducibility Challenge. In successfully reproducing the most crucial part of the paper, we reach a performance that is comparable with or superior to the original paper on two benchmarks for several settings. We evaluate new baseline results, using a new dataset presented in the paper. Yet, we also provide multiple remarks and recommendations about reproducibility and comparability. After we brought our reproducibility work to the authors' attention, they have updated the original paper on which this work is based and released code as well. Our contributions mainly consist in reproducing the most important results of their original paper, in giving insight in the reproducibility and in providing a first open-source implementation.

## 1   Introduction

The ability to adapt to new situations and learn quickly is a cornerstone of human intelligence. When given a previously unseen task, humans can use their previous experience and learning abilities to perform well on this new task in a matter of seconds and with a relatively small amount of new data. Artificial learning methods have been shown to be very effective for specific tasks, often times surpassing human performance (Silver et al. [2016], Esteva et al. [2017]). However, by relying on standard supervised-learning or reinforcement learning training paradigms, these artificial methods still require much training data and training time to adapt to a new task.

An area of machine learning that learns and adapts from a small amount of data is called *few-shot learning*. A *shot* corresponds to a single example, e.g. an image and its label. In few-shot learning the learning scope is expanded to a variety of tasks with a few shots each, compared to the classic setting of a single task with many shots. A promising approach for few-shot learning is the field of *meta-learning*. Meta-learning, also known as *learning-to-learn*, is a paradigm that exploits cross-task information and training experience to perform well on a new unseen task.

In this work we reproduce the paper of Bertinetto et al. [2019] (referenced as *"their paper"*); it falls into the class of gradient-based meta-learning algorithms that learn a model parameter intialization for rapid fine-tuning with a few shots (Finn et al. [2017], Nichol and Schulman [2018]). The authors present a new meta-learning method that combines a deep neural network feature extractor with differentiable learning algorithms that have closed-form solutions. This reduces the overall complexity of the gradient based meta-learning process, while advancing the state-of-the-art in terms of accuracy across multiple few-shot benchmarks.

We interacted with the authors through OpenReview[1], bringing our reproducibility work and Tensor-Flow code[2,3] to their attention. Because of this, they have recently updated their original paper with more details to facilitate reproduction and they have released an official PyTorch implementation[4].

## 2   Background in meta-learning

The objective of few-shot meta-learning is to train a model that can quickly adapt to a new task by using only a few datapoints and training iterations. In our work we will consider only classification tasks, but it should be noted that meta-learning is also generally applicable to regression or reinforcement learning tasks (Finn et al. [2017]).

In order to provide a solid definition of meta-learning, we need to define its different components. We denote the set of tasks by $\mathbb{T}$. A task $\mathcal{T}_i \in \mathbb{T}$ corresponds to a classification problem, with a probability distribution of example inputs $\boldsymbol{x}$ and (class) labels $y$, $(\boldsymbol{x}, y) \sim \mathcal{T}_i$. For each task, we are given training samples $\mathcal{Z}_{\mathcal{T}} = \{(\boldsymbol{x}_i, y_i)\} \sim \mathcal{T}$ with $K$ shots per class and evaluation samples $\mathcal{Z}'_{\mathcal{T}} = \{(\boldsymbol{x}'_i, y'_i)\} \sim \mathcal{T}$ with $Q$ shots (*queries*) per class, all sampled independently from the same distribution $\mathcal{T}$. In meta-learning, we reuse the learning experience used for tasks $\mathcal{T}_i$, $i \in [0, L]$ to learn a new task $\mathcal{T}_j$, where $j > L$, from only $K$ examples, for every single one of the $N$ classes in the task. Commonly, this is denoted as an *N-way K-shot* problem. To this end, in meta-learning two different kinds of learners can be at play: (1) a *base-learner* that works at the task level and learns a single task (e.g. classifier with $N$ classes) and (2) a *meta-learner* that produces those model parameters that enable the fastest average fine-tuning (using the *base-learner*) on unseen tasks.

The authors put a specific view of meta-learning forward. Their meta-learning system consists of a generic feature extractor $\Phi(\boldsymbol{x})$ that is parametrized by $\omega$, and a task-specific predictor $f_{\mathcal{T}}(X)$ that is parametrized by $w_{\mathcal{T}}$ and adapts separately to every task $\mathcal{T} \in \mathbb{T}$ based on the few shots available. In the case of a deep neural network architecture, this task-specific predictor $f_{\mathcal{T}}$ can be seen as the last layer(s) of the network and is specific to a task $\mathcal{T}$. The preceding layers $\Phi$ can be trained across tasks to provide the best feature extraction on which the task-specific predictor can finetune with maximum performance.

The base-learning phase in their paper assumes that the parameters $\omega$ of the feature extractor $\Phi$ are fixed and computes the parameters $w_{\mathcal{T}}$ of $f_{\mathcal{T}}$ through closed-form learning process $\Lambda$. $\Lambda$, on its own, is parametrized by $\rho$. The meta-learning phase in the paper learns a parametrization of $\Phi$ and $\Lambda$ (respectively $\omega$ and $\rho$). In order to learn those meta-parameters, the algorithm minimizes the expected loss on test sets from unseen tasks in $\mathbb{T}$ with gradient descent. The base-learning and meta-learning phases are shown in Figures 1 and 2, respectively.

Most of the recent meta-learning works are tested against image datasets and their feature extractor consists of a convolutional neural network (CNN). The variability between works resides mainly in the base learner $f_{\mathcal{T}}$ and its parameter obtaining training procedure $\Lambda$. Examples are an (unparametrized) k-nearest-neighbour algorithm (Vinyals et al. [2016]), a CNN with SGD (Mishra et al. [2017], and a nested SGD (Finn et al. [2017]). Systems in Vinyals et al. [2016] and Snell et al. [2017] are based on comparing new examples in a learned metric space and rely on matching. In particular, MATCHINGNET from Vinyals et al. [2016] uses neural networks augmented with memory and recurrence with attention in a few-shot image recognition context. Mishra et al. [2017] build on this attention technique by adding temporal convolutions to reuse information from past tasks. Another example of a matching-based method is introduced in Garcia and Bruna [2017], where a graph neural network learns the correspondence between the training and testing sets. A different approach is to consider the SGD update as a learnable function for meta-learning. In particular, sequential learning algorithms, such as recurrent neural networks and LSTM-based methods, enable the use of long-term dependencies between the data and gradient updates as pointed out by Ravi and Larochelle [2017]. Finally, Finn et al. [2017] introduce a technique called model-agnostic meta-learning (MAML). In MAML, meta-learning is done by backpropagating through the fine-tuning stochastic gradient descent update of the model parameters.

---

[1] `https://openreview.net/forum?id=HyxnZh0ct7&noteId=BkxDPnDZMV`

[2] our R2D2 and R2D2*: `https://github.com/ArnoutDevos/r2d2`

[3] our MAML on CIFAR-FS: `https://github.com/ArnoutDevos/maml-CIFAR-FS`

[4] Bertinetto et al. [2019] code: `https://github.com/bertinetto/r2d2`

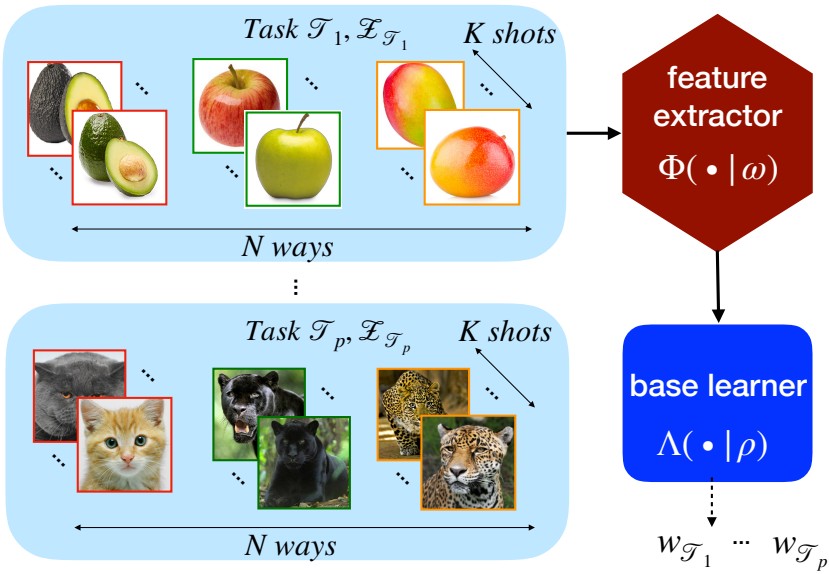

Figure 1: Base-learning of the task-specific parameters $w_{\mathcal{T}_i}$ over $p$ tasks following steps 3 to 6 of Algorithm 1.

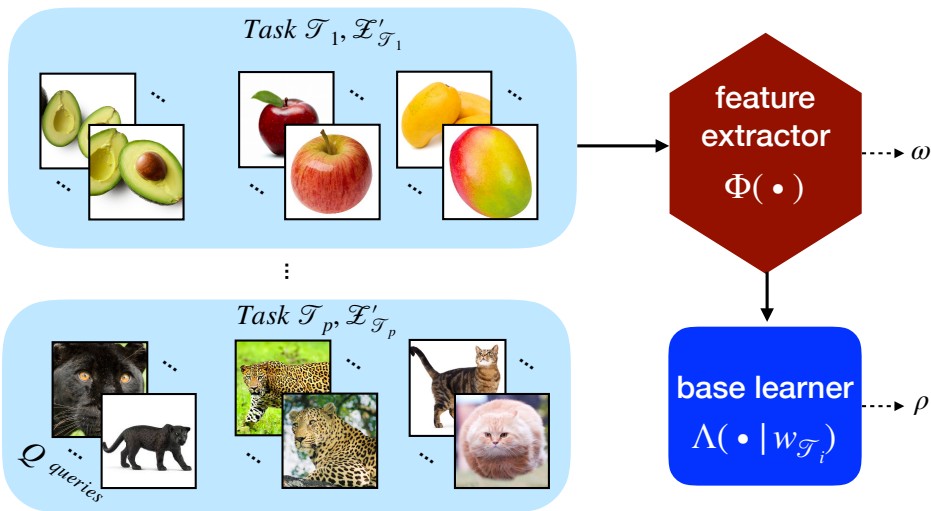

Figure 2: Meta-learning of the meta-parameters $\omega$ and $\boldsymbol{\rho}$ over the evaluation sets of each task $\mathcal{Z}'_{\mathcal{T}_i}$ using the previously learned $w_{\mathcal{T}_i}$ following steps 7 to 9 of Algorithm 1.

# 3 Analysis of the R2D2 Classifier

In their paper, Bertinetto et al. [2019] present a new approach that relies on using fast and simple base learners such as *ridge regression differentiable discriminator* (R2D2) or *(regularized) logistic regression differentiable discriminator* (LRD2). In our reproducibility work we will focus on the R2D2 algorithm, because it is the only proposed algorithm with a truly closed-form solver for the base-learner. For reproducibility purposes, we transformed the original textual description of R2D2 in their paper into an algorithmic description in Algorithm 1, elaborated upon in the following.

---

**Algorithm 1** Ridge Regression Differentiable Discriminator (R2D2)

---

**Require:** Distribution of tasks $\mathbb{T}$.
**Require:** Feature extractor $\Phi$ parameterized by $\omega$.
**Require:** Finetuning predictor $f_{\mathcal{T}}$ with base-learning algorithm $\Lambda$ and task-specific parameters $w_{\mathcal{T}}$, and meta-parameters $\boldsymbol{\rho} = (\alpha, \beta, \lambda)$
1: Initialize $\Phi$, $\Lambda$, and $f_{\mathcal{T}}$ with pre-trained or random parameters $\omega_0$ and $\boldsymbol{\rho_0}$
2: **while** not done **do**
3:      Sample batch of tasks $\mathcal{T}_i \sim \mathbb{T}$
4:      **for all** $\mathcal{T}_i$ **do**
5:          Sample $K$ datapoints for every class from $\mathcal{T}_i$ and put in them in the *training set* $\mathcal{Z}_{\mathcal{T}_i}$
6:          Base-learn $f_{\mathcal{T}_i}$ using $\Lambda$:
         $W_i = w_{\mathcal{T}_i} = \Lambda(\mathcal{Z}_{\mathcal{T}_i}) = X_i^T(X_i X_i^T + \lambda.I)^{-1}Y_i$
         with $X_i = \Phi(\mathcal{Z}_{\mathcal{T}_i})$ and $Y_i$ the one-hot labels from $\mathcal{Z}_{\mathcal{T}_i}$.

7:          Sample datapoints for every class from $\mathcal{T}_i$ and put in them in the *evaluation set* $\mathcal{Z}'_{\mathcal{T}_i}$
8:      **end for**
9:      Update meta-parameters $\theta = (\omega, \boldsymbol{\rho})$ through gradient descent :

$$\theta \leftarrow \theta - \varepsilon . \sum_i \nabla_\theta \mathcal{L}(f_{\mathcal{T}_i}(\Phi(\mathcal{Z}'_{\mathcal{T}_i})), Y'_i)$$

     with $\varepsilon$ the learning rate, $\mathcal{L}$ the cross-entropy loss, and $f_{\mathcal{T}_i}(X'_i) = \alpha X'_i W_i + \beta$.
10: **end while**

---

In R2D2, during base-learning with $\mathcal{Z}_{\mathcal{T}}$, the linear predictor $f_{\mathcal{T}}$ is adapted for each training task $\mathcal{T}$, by using the learning algorithm $\Lambda$; and the meta-parameters $\omega$ (of $\Phi$) and $\boldsymbol{\rho}$ (of $\Lambda$) remain fixed. It is only in the meta-training phase that meta-parameters $\omega$ and $\boldsymbol{\rho}$ are updated, by using $\mathcal{Z}'_{\mathcal{T}}$. The linear predictor is seen as $f_{\mathcal{T}}(x) = xW$ with $W$ a matrix of task-specific weights $w_{\mathcal{T}}$, and $x$ the feature extracted version of $\boldsymbol{x}$, $x = \Phi(\boldsymbol{x})$. This approach leads to a ridge regression evaluation such that it learns the task weights $w_{\mathcal{T}}$:

$$\Lambda(X, Y) = \underset{W}{\arg\min} \|XW - Y\|^2 + \lambda \|W\|^2 \tag{1}$$

$$= (X^T X + \lambda I)^{-1} X^T Y \tag{2}$$

where $X$ contains all $NK$ feature extracted inputs from the training set of the considered task. A key insight in their paper is that the closed-form solution of Equation 2 can be simplified using the *Woodbury matrix identity* yielding $W = \Lambda(X, Y) = X^T(XX^T + \lambda I)^{-1}Y$. This considerably reduces the complexity of the matrix calculations in the special case of few-shot learning. Specifically, $XX^T$ is of size $NK \times NK$, in the case of an $N$-way $K$-shot task; this matrix will, together with the regularization, be relatively easily inverted. Normally, regression is not adequate for classification, but the authors noticed that it still has considerable performance. Therefore, in order to transform the regression outputs (which are only effectively calculated when updating the meta-parameters using $\mathcal{Z}'_{\mathcal{T}}$) to work with the cross-entropy loss function, the meta-parameters $(\alpha, \beta) \in \mathbb{R}^2$ serve as a scale and bias, respectively:

$$\hat{Y}' = \alpha X' \left[ X^T(XX^T + \lambda I)^{-1}Y \right] + \beta \tag{3}$$

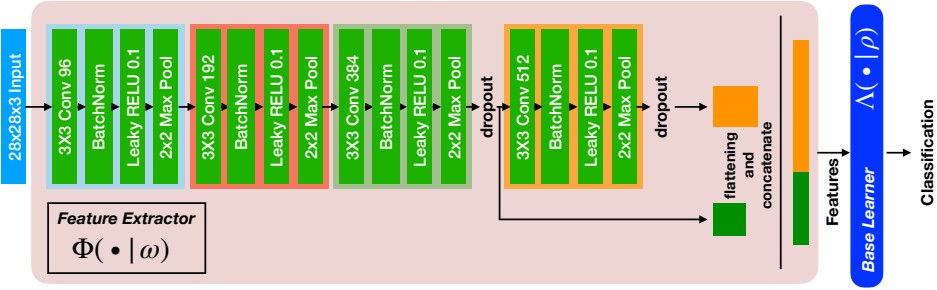

Figure 3: Overall architecture of the R2D2 system considering [96, 192, 384, 512] filters in the feature extractor with 4 convolutional blocks for the CIFAR-FS dataset.

## 4 Reproducibility

As a first step in the reproducibility, we reproduce the results of a baseline algorithm on different datasets used in their paper. In this perspective, we first consider the MAML algorithm from Finn et al. [2017]. We use the official TensorFlow implementation of MAML (Finn [2018]) to reproduce the baseline's results. Then, we amend this MAML implementation to reproduce the results on the new CIFAR-FS dataset proposed by their paper (Bertinetto et al. [2019]).

When reproducing the R2D2 algorithm, our first consideration is that the feature extractors in MAML and R2D2 are very different. MAML uses four convolutional blocks with an organization of [32, 32, 32, 32] filters. Whereas, R2D2's four blocks employ a [96, 192, 384, 512] scheme, as shown in Figure 3. In other words, the feature extractor in R2D2 is more complex hence is expected to yield better results (Mhaskar et al. [2016]). In order to provide a meaningful comparison, we implement and evaluate both the simple and more complex feature extractors for the R2D2 algorithm, denoted by R2D2* and R2D2 respectively.

In order to make a working reproduction of their paper we had to make the following assumptions. We first considered the aforementioned complex architecture and feature extractor. In particular, for the feature extractor, we made assumptions on the convolutional block options. We considered a 3x3 convolution block with a 'same' padding and a stride of 1. For the 2x2 maximum pooling, we use a stride of 2 and no padding. Second, concerning the ridge regression base-learner, we opted for a multinomial regression that returns the class with the maximum value through one-hot encoding. Following the guidelines for the feature extractor presented in Section 4.2 of their paper, we were not successful in reproducing the exact number of features at the output of the feature extractor. In their paper, the overall numbers of features at the output of the extractor are 3584, 72576 and 8064 for Omniglot, miniImageNet and CIFAR-FS, respectively. However, by implementing the feature extractor described in their paper, we obtain 3988, 51200 and 8192 respectively.

For comparison purposes, we use the same number of classes (e.g. 5) and shots during (e.g. 1) training and testing, despite their paper using a higher number of classes during training (16 for miniImageNet, 20 for CIFAR-FS) than during testing (5 for miniImageNet and CIFAR-FS). Regarding the amount of shots, their paper uses a random number of shots during training. This is different from the way most baselines are trained using the same number of shots per class during training and testing (Finn [2018], Nichol and Schulman [2018], Vinyals et al. [2016]). For comparability, it is paramount to keep the training and testing procedures similar, if not the same. In particular, as in their paper the 5-way results are exactly the same as those reported in MAML (Finn et al. [2017]), using the same number of classes and shots during training and testing allows for a justifiable comparison.

Finally, a last assumption is made on the algorithm's stopping criterion. In their paper, the stopping criterion is vaguely defined as "*the error on the meta-validation set does not decrease meaningfully for 20,000 episodes*". Therefore, in line with the MAML training procedure, we meta-train using 60,000 iterations. To update the meta-parameters, in line with their paper, we use the Adam optimizer (Kingma and Ba [2014]) with an initial learning rate of 0.005, dampened by 0.5 every 2,000 episodes. We use 15 examples per class for evaluating the post-update meta-gradient. We use a meta batch-size of 4 and 2 tasks for 1-shot and 5-shot training respectively. For MAML we use a task-level learning rate of 0.01, with 5 steps during training and 10 steps during testing.

# 5 Results and contributions

The results of the different implemented architectures and algorithms for several datasets are shown in Figures 4 and 5. More detailed results with 95% confidence intervals are shown in Tables 1 and 2. The first and last column correspond to the baselines in original papers.

Our implementations were made in Python 3.6.2 and TensorFlow 1.8.0 (Abadi et al. [2016]). The source code of all implementations is available[5] online[6]. The simulations were run on a machine with 24 Xeon e5 2680s at 2.5 GHz, 252GB RAM and a Titan X GPU with 12 GB RAM.

Although our results differ slightly from the original paper of Bertinetto et al. [2019], R2D2 (with its more complex network architecture) performs better than the MAML method for most simulations. It is not a surprise that, in most of the cases, with a more complex feature extractor better results are obtained for the same algorithm (R2D2 vs R2D2*). Overall, our study confirms that the R2D2 meta-learning method, with its corresponding complex architecture, yields better performance than basic MAML (with its simpler architecture). The differences between reproduced results and reported values might be due to our assumptions or the stopping criterion in the training. Also, as expected, the complexity (N-ways) and the amount of data (K-shots) play a major role in the classification accuracy. The accuracy drops when the number of ways increases and number of shots decreases. An outlier worth mentioning is our MAML simulation on miniImageNet: the 2-way 1-shot classification accuracy of $78.8 \pm 2.8\%$ is much better than the $74.9 \pm 3.0\%$ reported in Finn et al. [2017].

In summary, we successfully reproduced the most important results presented in Bertinetto et al. [2019]. Although our reproduced results and their paper results differ slightly, the general observations of the authors remain valid. Their meta-learning with differentiable closed-form solvers yields state-of-the-art results and improves over another state-of-the-art method. The assumptions made, however, could have been clarified in their original paper. Indeed, these assumptions could be the source of the discrepancy in the reproduction results. In this reproducibility work we did not focus on the logistic regression based algorithm (LRD2) from their paper because the logistic regression solver does not have a closed-form solution.

Overall, with this reproducibility project we make the following contributions:

- Algorithmic description of the R2D2 version of meta-learning with differentiable closed-form solvers (Algorithm 1).
- Evaluation of the MAML pipeline from Finn [2018] on two datasets: the existing miniImageNet and new CIFAR-FS for different few-shot multi-class settings.
- Implementation of R2D2* in TensorFlow on the pipeline following Algorithm 1 with the original MAML feature extractor.
- Implementation of R2D2 in TensorFlow on the pipeline following Algorithm 1 with the Figure 3 architecture as mimicked from in the original paper (Bertinetto et al. [2019]).
- Evaluation and insights in the reproducibility of Bertinetto et al. [2019].

# 6 Conclusion

In this work we have presented a reproducibility analysis of the ICLR 2019 paper *"Meta-learning with differentiable closed-form solvers"* by Bertinetto et al. [2019]. Some parameters and training methodologies, which would be required for full reproducibility, such as *stride* and *padding* of the convolutional filters, and a clear stopping criterion, are not mentioned in the original paper or in its appendix (Bertinetto et al. [2019]). However, by making reasonable assumptions, we have been able to reproduce the most important parts of the paper and to achieve similar results. Most importantly we have succeeded in reproducing the increase in performance of the proposed method over some reproduced baseline results, which supports the conclusions of the original paper. However, the different neural network architectures should be taken into consideration when comparing results.

---

[5]R2D2 and R2D2*: `https://github.com/ArnoutDevos/r2d2`
[6]MAML with CIFAR-FS: `https://github.com/ArnoutDevos/maml-CIFAR-FS`

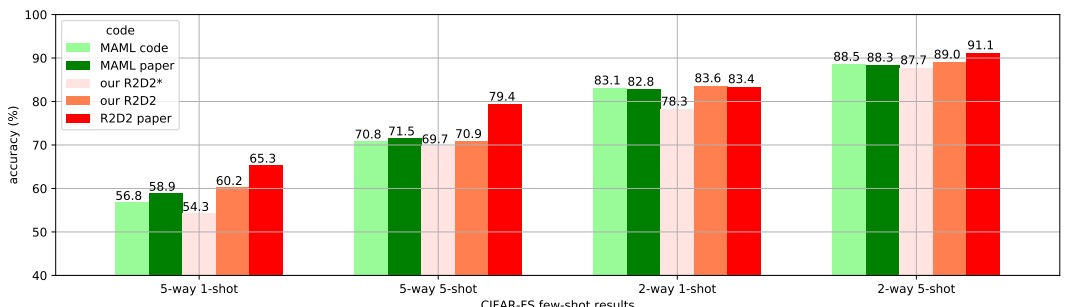

Figure 4: $N$-way $K$-shot classification accuracies on CIFAR-FS. Detailed results in Table 1.

Table 1: $N$-way $K$-shot classification accuracies on CIFAR-FS with 95% confidence intervals.

| Method | MAML paper Bertinetto et al. [2019] | MAML ours | R2D2* ours | R2D2 ours | R2D2 paper Bertinetto et al. [2019] |
|---|---|---|---|---|---|
| 5-way, 1-shot | $58.9 \pm 1.9\%$ | $56.8 \pm 1.9\%$ | $54.3 \pm 1.8\%$ | $60.2 \pm 1.8\%$ | $65.3 \pm 0.2\%$ |
| 5-way, 5-shot | $71.5 \pm 1.0\%$ | $70.8 \pm 0.9\%$ | $69.7 \pm 0.9\%$ | $70.9 \pm 0.9\%$ | $79.4 \pm 0.1\%$ |
| 2-way, 1-shot | $82.8 \pm 2.7\%$ | $83.1 \pm 2.6\%$ | $78.3 \pm 2.8\%$ | $83.6 \pm 2.6\%$ | $83.4 \pm 0.3\%$ |
| 2-way, 5-shot | $88.3 \pm 1.1\%$ | $88.5 \pm 1.1\%$ | $87.7 \pm 1.1\%$ | $89.0 \pm 1.0\%$ | $91.1 \pm 0.2\%$ |

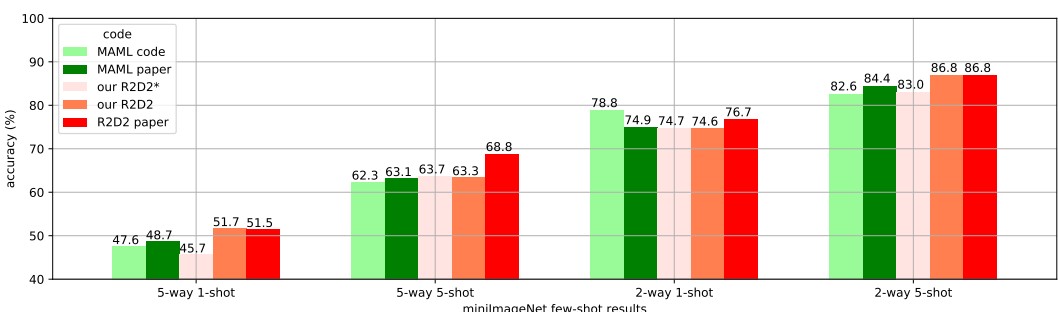

Figure 5: $N$-way $K$-shot classification accuracies on miniImageNet. Detailed results in Table 2.

Table 2: $N$-way $K$-shot classification accuracies on miniImageNet with 95% confidence intervals

| Method | MAML paper Finn et al. [2017] | MAML code Finn [2018] | R2D2* ours | R2D2 ours | R2D2 paper Bertinetto et al. [2019] |
|---|---|---|---|---|---|
| 5-way, 1-shot | $48.7 \pm 1.8\%$ | $47.6 \pm 1.9\%$ | $45.7 \pm 1.8\%$ | $51.7 \pm 1.8\%$ | $51.5 \pm 0.2\%$ |
| 5-way, 5-shot | $63.1 \pm 0.9\%$ | $62.3 \pm 0.9\%$ | $63.7 \pm 1.3\%$ | $63.3 \pm 0.9\%$ | $68.8 \pm 0.2\%$ |
| 2-way, 1-shot | $74.9 \pm 3.0\%$ | $78.8 \pm 2.8\%$ | $74.7 \pm 2.9\%$ | $74.6 \pm 2.9\%$ | $76.7 \pm 0.3\%$ |
| 2-way, 5-shot | $84.4 \pm 1.2\%$ | $82.6 \pm 1.2\%$ | $83.0 \pm 1.2\%$ | $84.6 \pm 1.2\%$ | $86.8 \pm 0.2\%$ |

## Acknowledgements

The authors would like to thank Martin Jaggi, Ruediger Urbanke, and the anonymous reviewers from the ICLR 2019 Reproducibility Challenge for feedback. This project is partially supported by the European Union's Horizon 2020 research and innovation program under the Marie Skłodowska-Curie grant agreement No. 754354.

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
