# OpenReview forum: "Reproducing Meta-learning with differentiable closed-form solvers"
_ICLR.cc/2019/Workshop/RML — RML 2019_

### Official Review · AnonReviewer1 · 2019-04-01
**Interesting Work**

**Rating:** 5
**Confidence:** 3

**Review:**

Summary: This paper is a reproduction attempt of the Bertinetto et al. 2019 paper “Meta-learning with differentiable closed-form solvers”, an improvement over Finn’s 2018 MAML (Model-Agnostic Meta-Learning) paper. It is a competently-executed reproduction attempt. It has successfully reproduced Bertinetto et al, modulo small, insignificant differences that do not affect the conclusions, provides a TensorFlow implementation of the original authors’ proposed algorithm, and additionally resulted in the original authors releasing a PyTorch implementation.

The reproduction focuses on Bertinetto’s proposed Ridge-Regression Differentiable Discriminator (R2D2), leaving aside their Logistic Regression Differentiable Discriminator (LRD2) because its solution is not truly closed-form. The reproduction provides a service to readers of Bertinetto’s paper by providing the R2D2 algorithm in pseudo-code, rather than textual form.

The reproduction provides some background context for Bertinetto’s paper, explaining the N-way K-shot (few-shot) meta-learning problem. Figures 1 & 2, however, do not succeed in explaining the problem well; They are as helpful as they are confusing.

Whereas Bertinetto’s paper fails to provide certain parameters of the neural network’s convolutions layers, among others, the reproduction attempt correctly guesses and publishes them with their reproduction source code. Correspondence with Bertinetto et al via OpenReview has led to improvements to their paper and release of their code.

The reproduction attempt strengthens in some places Bertinetto’s paper (and the older MAML paper), and is a worthy contribution to the workshop and to science in general. My only comment is that Figures 1 & 2 are not clear, and distract from the background explanation.

---

### Decision · Program_Chairs · 2019-04-05
**Acceptance Decision**

Accept